# Response Surface Methodology Using Observational Data: A Systematic Literature Review

**Mochammad Arbi Hadiyat** [1,2,*] **, Bertha Maya Sopha** [1] **and Budhi Sholeh Wibowo** [1]

1   Industrial Engineering Program, Department of Mechanical and Industrial Engineering, Universitas Gadjah Mada, Yogyakarta 55281, Indonesia
2   Industrial Engineering Program, Faculty of Engineering, Universitas Surabaya (Ubaya), Surabaya 60293, Indonesia
*   Correspondence: arbi@mail.ugm.ac.id or arbi@staff.ubaya.ac.id

**Abstract:** In the response surface methodology (RSM), the designed experiment helps create interfactor orthogonality and interpretable response models for the purpose of process and design optimization. However, along with the development of data-recording technology, observational data have emerged as an alternative to experimental data, and they contain potential information on design/process parameters (as factors) and product characteristics that are useful for RSM analysis. Recent studies in various fields have proposed modifications to the standard RSM procedures to adopt observational data and attain considerable results despite some limitations. This paper aims to explore various methods to incorporate observational data in the RSM through a systematic literature review. More than 400 papers were retrieved from the Scopus database, and 83 were selected and carefully reviewed. To adopt observational data, modifications to the procedures of RSM analysis include the design of the experiment (DoE), response modeling, and design/process optimization. The proposed approaches were then mapped to capture the sequence of the modified RSM analysis. The findings highlight the novelty of observational-data-based RSM (RSM-OD) for generating reproducible results involving the discussion of the treatments for observational data as an alternative to the DoE, the refinement of the RSM model to fit the data, and the adaptation of the optimization technique. Future potential research, such as the improvement of factor orthogonality and RSM model modifications, is also discussed.

**Keywords:** classic RSM; observational data; RSM-OD; RSM stages; systematic literature review

## 1. Introduction

Since first introduced by Box and Wilson in 1951 [1], response surface methodology (RSM) has been widely used by scientists and engineers to find optimal parameter settings to improve a process and equipment designs. The RSM adopts the design of experiment (DoE) concept to collect data and identify significant factors and interactions that influence the process response. Next, RSM is used to develop a mathematical model to capture the causal relationships between factors and responses. Thus, the final result of RSM is obtaining optimal factor settings by optimizing the causality model as the objective function. As one of the common techniques for process optimization, this method works in situations where engineers have complete control over the factor levels and treatments, such as in laboratory experiments, scientific method applications, computer experiments, and any other research environments that involve controllable factors. For certain industrial processes or design optimization, RSM provides a means for engineers to find the best parameter settings to optimize process/product characteristics. As long as engineers have the chance to set the process/equipment parameter, then the RSM can ideally work based on experimentation activities.

Nevertheless, conducting designed experiments for continuous process/production is challenging. Changing the parameters during the running process can disturb production,

and increase the number of nonconforming items, hence, resulting in high costs [2]. When direct experimentation is not feasible, one of the alternative solutions is to use observational data as the input to the RSM [3,4]. Some high-tech industries are often complemented with intelligent data-acquiring systems that allow them to record real-time process/equipment parameter changes. For prediction purposes, these real-time recorded data become the input for a mathematical model to generate outputs, such as a forecasting system for maintenance schedules or product quality [5]. Several pieces of research on chemical engineering and food production [6,7] have demonstrated that observational-data-based RSM (RSM-OD) provided a fitted mathematical model to find an optimal factor setting. Other research used the observed data from a running process or equipment as the input for RSM-OD, as shown in the work of [8] for steel production and [9] with pollutant removal processes.

However, observational data and their similarities, including real-time recording data and already conducted experiment data, limits a researcher's control over their factor levels, as the DoE ideally affords. There are presumptions that observational data contain a high volume, high variability, unstructured, and serial-correlated situations [10]. Therefore, some modifications to selecting the observations are required prior to the use of the data in RSM analysis, including the adaptive RSM model and optimization techniques, while still considering the ideal concept of the RSM. The authors of [3] have successfully adopted observational data for the DoE by selecting a subset of observations and identifying stages within the data, similarly, Refs. [2,11,12] also giving alternatives by matching the data with certain DoE to ensure orthogonality. Moreover, the authors of [4,13] apply the RSM to real-time data acquisition for optimization during continuous processes. It is also worth noting that the recent development of big data has accelerated the use of observational data. For instance, Ref. [14] demonstrated real-manufacturing-oriented big data, in which recorded datasets provide information for process improvement and optimization. The data-recording technology provides massive datasets in which huge datasets are recorded along with operations [15–17]. Once the acquired dataset contains the process parameters and product characteristics, then the RSM-OD should be considered as an optimization methodology. However, existing pieces of literature on RSM-OD have a unique approach to treating the observational data and modifying the RSM model or procedures; thus, the opportunities to develop an established RSM-OD are still open.

Therefore, the paper aims to explore various approaches to develop RSM-OD through a systematic literature review. The review was based on 82 pieces of literature which were selected and analyzed using the PRISMA framework [18]. The paper focuses on how observational data can be considered as input for RSM for process/design optimization purposes. According to the authors' best knowledge, the present paper is the first comprehensive review of the successful implementation of RSM-OD in various research fields. Other review studies on RSM systematic literature review papers have discussed classic RSM and DoE in advanced manufacturing optimization [19] and neural network, replacing the DoE model [20]. Hence, the paper contributes by providing insights into the development of new procedures in RSM-OD following three stages of analysis in standard classic RSM, i.e., the treatment of nondesigned experimental data, the modeling of the relationship between factors and response, and optimization.

The rest of the paper is structured as follows. Section 2 briefly explains how the classic and ideal RSM model works based on experimental data and the opportunity to adopt observational data. Section 3 describes the systematic literature review (SLR) methodology. Section 4 presents the results of descriptive and bibliometric analysis, which is followed by synthesis and discussion in Section 5. Lastly, Section 6 concludes by highlighting the main findings, limitations, and future research.

## 2. RSM Overview for Response Optimization

This section contains a theoretical perspective of the classic RSM and its applications in various research fields. Considerable research on the classic RSM showed that this

method has recently provided significant contributions. A designed experiment-based RSM with fulfilled statistical assumptions will give strong theoretically-based analysis and interpretation. Nevertheless, the consideration of using observational data in RSM should not be ignored because some pieces of literature [4,13] have demonstrated the successful implementation of observational data in the RSM. The section also presents some of these papers as motivating examples of the rationale for writing this paper.

### 2.1. Classic RSM

As mentioned above, classical RSM works by integrating three tools in a sequential analysis (Figure 1). In the first stage, classic RSM implements the DoE. In this step, the DoE plays a role in experiment planning, data collection, analysis, and interpretation and ensures that the experiment fulfills its purpose. Orthogonality fulfillment in the DoE matrix ensures that the predetermined process parameters can be estimated independently among others. Second, classic RSM applies a specific mathematical model to fit the data obtained by the DoE. This model captures the relationship between factors or parameters as inputs and responses as outputs. Classic RSM usually prefers to adopt a linear model because of its simple interpretation and formal statistical inference of all its required assumptions during the modeling stage. Third, the optimization stage works by finding the factor (or parameter) setting to optimize the response. Standard optimization tools, such as mathematical optimization and desirability functions [21], are preferred in classic RSM, along with some theoretical approaches. As the required assumptions in RSM are fulfilled for each stage, this methodology has become the best choice rather than any modification.

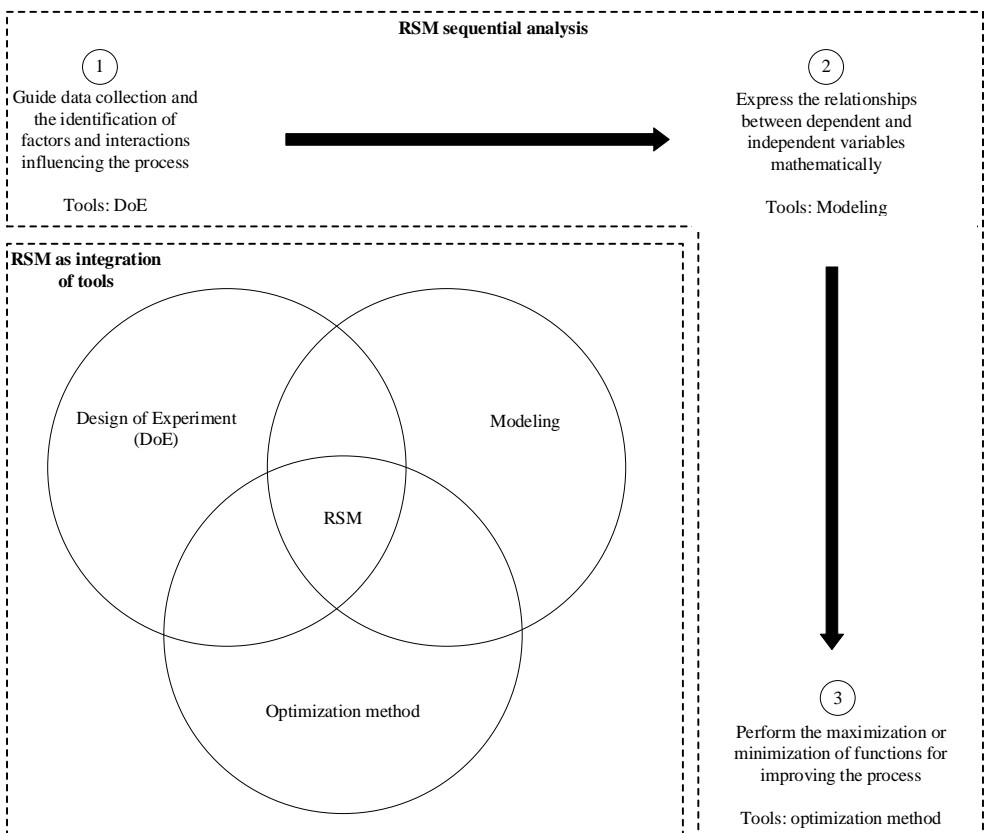

**Figure 1.** Overview of RSM (adopted from [19]).

In addition, an essential prestage in RSM involves determining the factors involved in the analysis. As the DoE is applied, researchers should subjectively select the factors in RSM. They need to find the factors with more minor or significant effects on the response based on previous research, considering the scope and knowledge domain of the researchers.

As an established methodology for designed, experimentally-based optimization, the classic RSM has been successfully applied for years in many research fields. Starting with the concept proposed by [22], more than 48,000 Scopus-indexed papers applied classic RSM. Figure 2 shows that the general engineering fields dominate the percentage of RSM applications, followed by chemical engineering, chemistry, biological sciences, and other applied sciences. It means that RSM plays an important role as an optimization methodology in many research fields, and there are also considerable developments in RSM to accommodate recent research issues.

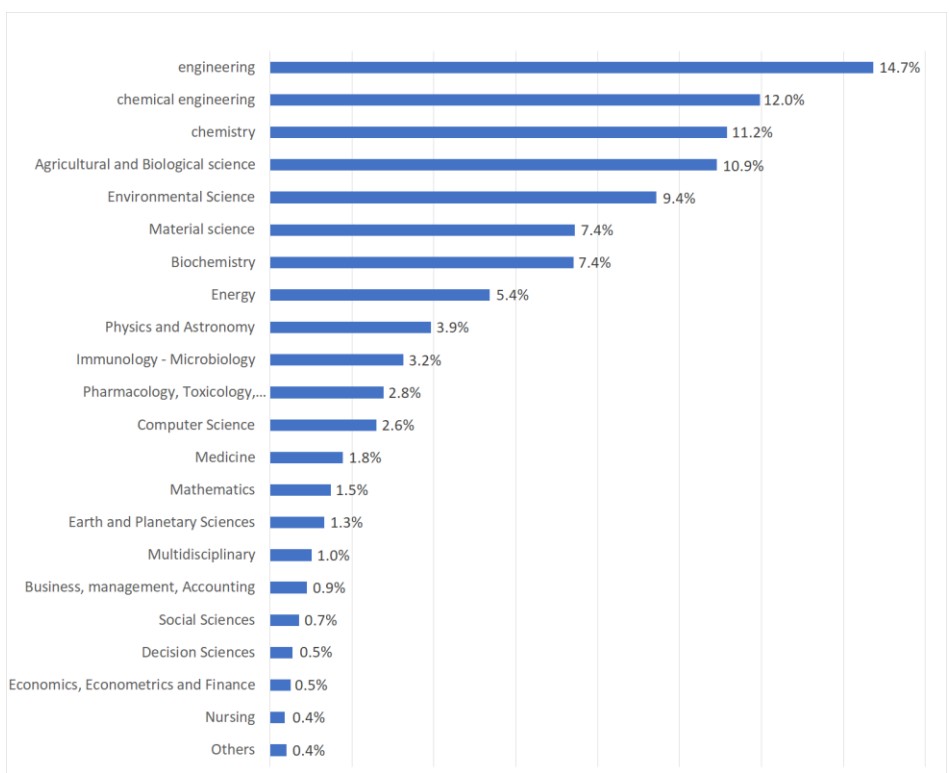

**Figure 2.** Research fields applying RSM.

Many improvements to classic RSM have been performed, mainly when optimization of the target by standard RSM procedures provides dissatisfactory results. Some papers on RSM improved the linear model to increase its performance in capturing the causality between factors and responses by replacing it with nonlinear versions. For example, [23,24] applied neural networks and support vector regression for RSM modeling to optimize surface roughness, respectively, in the milling and turning process. Other researchers [25] provided a similar approach that uses the RSM neural network model to optimize iron extraction from food. The complexity of these modified RSM models requires advanced optimization techniques and adopt a meta-heuristics method; for example, the authors of [26,27] successfully adopted a genetic algorithm for injection-molding and CNC process optimization.

Classic RSM can be improved by some modifications in order to enhance the performance of the process being investigated. However, all the methodological improvements of classic RSM should consider the basic concept of RSM, its stages, and the final purpose of the RSM, i.e., process optimization.

### 2.2. RSM-OD

The data-driven concept as a part of smart manufacturing has grown and has become a recent issue in some research, as proposed in other literature reviews [28,29]. Moreover, the rapid development of data acquisition systems supports the application of data-driven

analysis. In the manufacturing process, a data acquisition system, especially those with automatic sensor-based data recording, will produce massive mounts of data that potentially contain information about the characteristics of the process/equipment.

This system records data on the equipment parameters and product characteristics; as examples of the data-acquiring process, as explained by [5,15], some smart sensor devices can collect data from various types of equipment as a part of the data-driven technology. Therefore, several researchers argued that data analysis should be applied to obtain useful information. Other research successfully performed analyses based on these collected data for industrial application purposes, such as product quality prediction [30], preventive equipment maintenance [16], the process optimization purposes [4], similar to our topic. For practical purposes within manufacturing or laboratory scale, with the provided dataset or data acquiring system, the RSM-OD analysis is preferred because it does not need to interrupt the ongoing production, nor does it require exceptional equipment parameter adjustments for experimenting. Other papers argued that it reduces experimental costs [2].

Both sets of authors from [3,4] considered observational data as alternatives to designing experiments and applied them for continuous semiconductor and tire production, respectively. A large number of recorded data opened up opportunities to use them as a part of the process optimization system based on the data-driven concept. Both research papers showed how the RSM concept incorporates observational or historical data as the basis for process optimization. Specific iterative procedures, such as the selection of potential factors, the identification of stages in the dataset, and the search for a subset of observations with similar characteristics to the designed experiment, were proposed to treat the dataset to become suitable to adopt RSM.

In addition, some papers with laboratory-scope experimentation implemented RSM based on observational or historical data with a specific approach called historical data design (HDD), which is provided by Design Expert® v.11 software from Stat-Ease, Inc., 1300 Godward St NE, Suite 6400, Minneapolis, MN, USA. Although it is more similar to ordinary multiple regression analysis fitted to observational data, HDD is a type of observational data-based DoE within the RSM analysis. For details on performing HDD, refer to the software manual guide from [31] based on a case study by [32]. Both [6,8] explicitly mentioned and applied HDD, where previous, un-designed, and experimental or observational data were used as inputs for RSM analysis to optimize energy consumption and plastic strength.

Another similar paper gave a different perspective; the authors of [33] worked on an additive manufacturing process to predict surface roughness, and real-time data-driven modeling techniques were applied to minimize the prediction error. A real-time approach requires no assumptions for the data and does not need to evaluate the significance of the factors; its main target is to obtain the minimum error in the predicted response with no model interpretation required [34]. Meanwhile, the standard RSM proposed by [1] applies the philosophy of three stages in its analysis (Figure 1), with several required assumptions in the data, such as factor independence, treatment randomization, and factor significance, to give a strong interpretation; the final target of this RSM is to obtain the optimum response by finding the optimal factor setting/level.

The next section explains that this approach treats the dataset's variables, features, and responses as input and output. Some papers provided additional filtering procedures for selecting available observations to be fitted in the RSM model by treating the dataset so as to become similar to the designed experimental data [3,35]. Moreover, they applied machine learning models, such as neural networks and support vector machines (SVM), to replace ordinary linear models. However, this action will increase the risk of black-box modeling rather than keep the concept of model interpretability.

A number of RSM modifications to accommodate observational data have been conducted. Some modifications focused on data treatment before being used as input for the RSM. Other modifications develop adaptive mathematical modeling to any data condition,

including the use of machine learning approaches. The most recent modifications deal with the ability of optimization techniques to solve complex RSM models.

## 3. Methodology

The systematic literature review conducted in the present study follows systematic literature review guidance from [36] and conforms to the PRISMA statement in [18]. We started by identifying studies and followed this with database searches, filtering processes, and content analyses, as shown in Figure 3.

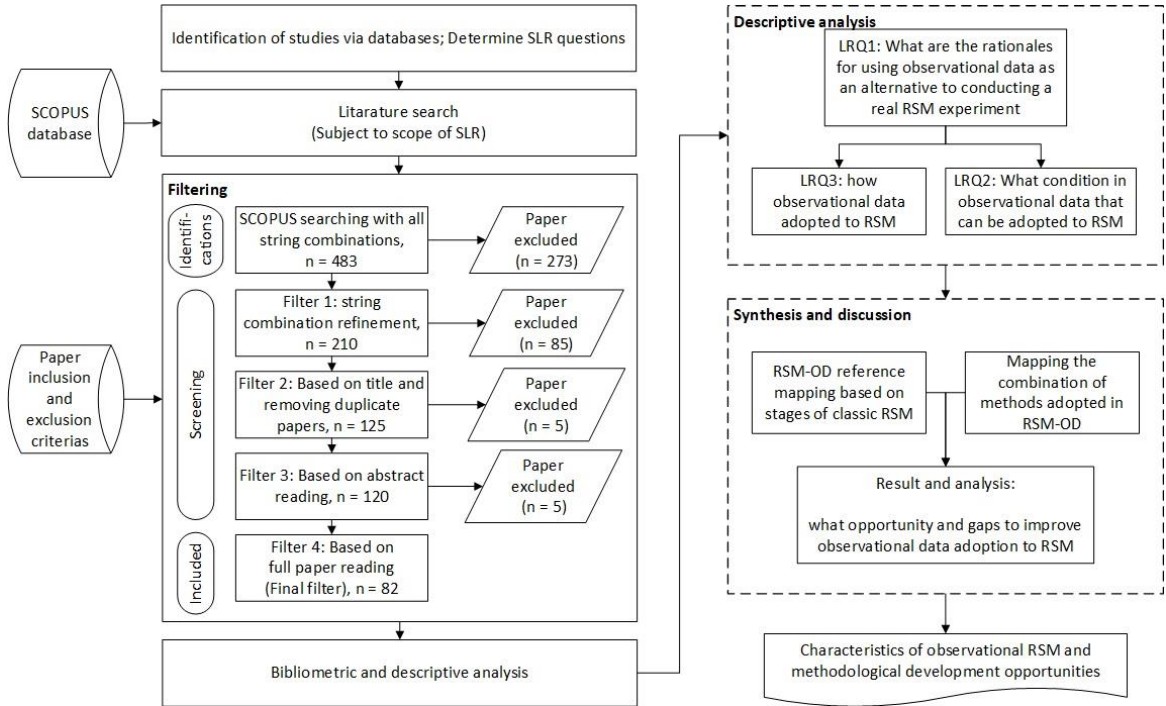

**Figure 3.** Systematic literature review framework based on PRISMA [18].

A systematic literature review gives an objective synthesis as it involves a decent number of references based on selected keywords. It follows the identification of studies, and the stages involve paper searches, filtering, and synthesis. As shown in Figure 3, identification steps define the problems in RSM, which are then formulated as research questions. By applying certain criteria based on the research questions, the collected pieces of literature were screened and analyzed with respect to descriptive, bibliometric, and comparative analysis. The Scopus database was deployed because it provided better article searching in terms of source titles, journal impact metrics, and the number of publishers when compared to others, as shown by [37].

The systematic literature review methodology was used to achieve a reproducible result in the development and application of RSM-OD. The analysis and discussion in this paper focused on those approaches accommodating nondesigned experimental data in the classic RSM. Moreover, as the context of this paper discusses the development of RSM-OD, the literature research questions (LRQs) emphasize how the standard RSM is modified to accept data.

- LRQ1: What are the rationales for using observational data as an alternative to conducting a real RSM experiment?
- LRQ2: What condition within observational data can be adapted to RSM?
- LRQ3: How are observational data adopted to RSM?

The descriptive analysis and synthesis stages in this paper attempted to answer those LRQs associated with the need for well-designed experimentally-based optimization in

various fields of studies. The practical limitations of conducting the experiments were raised and prompted the consideration of adopting observational data as an alternative. As shown in Figure 3, the stages start with a bibliometric analysis to map the interrelationship among research keywords as a reference for methodological mapping and answering the LRQs. LRQ1 was answered by identifying the rationales for using observational data for RSM analysis, considering the limitations of classic RSM in practice but still referring to its standard procedures (Figure 1). As LRQ1 was answered, LRQ2 and LRQ3 can parallelly be processed. The answers to LRQ2 review strict assumptions of the RSM and how observational data can still be adopted by RSM. As a result, observational data preprocessing and evaluations to conform to the RSM analysis were identified. Meanwhile, LRQ3 dealt with the procedures to adopt observational data into the RSM analysis, subject to the classic statistical assumptions within, including DoE properties, modified mathematical models, and the optimization method. Finally, the discussion was started based on the results of LRQ1 to LRQ3, which focus on the opportunities and gaps for adopting observational data as an alternative to DoE in RSM analysis and open the potential development of further research.

Search strings for paper abstracts and titles by restricting the search references from the Scopus database were predetermined to ensure that the papers still covered the proposed topics (Figure 4). Similar terms related to nonexperimental data used in the reference papers, such as observational, historical, or retrospective data, were found. Further, these terms were combined with common keywords in the RSM analysis, such as "optimization" and "DoE". Some Boolean operators were applied, considering that RSM-OD analysis should refer to the classic RSM stages.

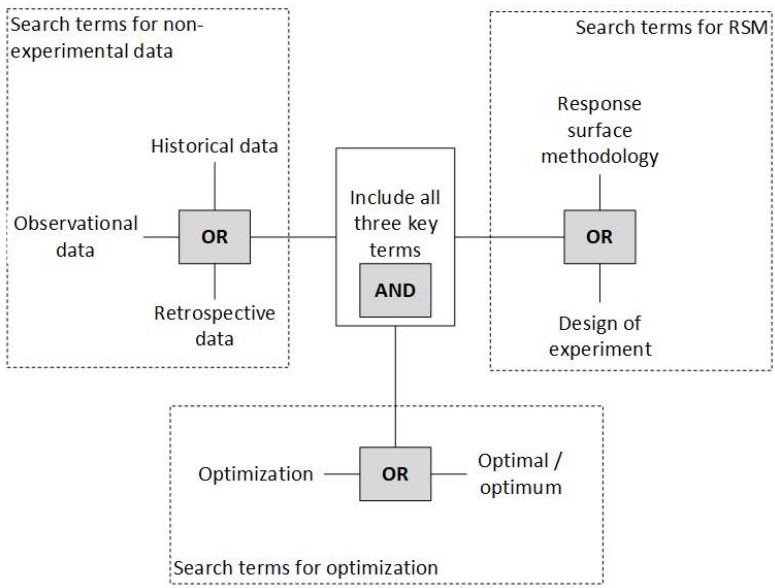

**Figure 4.** Search string and Boolean operators.

Figure 4 shows the search process. Based on the research questions, the key terms were "RSM", "non-experimental data", and "optimization". The search queries involved all of these, along with the use of the Boolean operator "AND". To enrich the search process, we identified some synonyms within each of the key terms based on the mentioned terms in reference papers. For example, several papers used different terms when mentioning the nonexperimental data but gave similar meanings, i.e., observational, historical, or retrospective data. One of these similar terms was then selected with the Boolean operator "OR" to complete the search string.

The search result with the determined search strings and Boolean operators yielded more than 400 papers from the SCOPUS database. However, not all of these papers discussed RSM with regard to observational data. Some mentioned similar keywords, but

the topics were outside of this paper's scope. The filtering process was then applied with the inclusion criteria in Table 1. The selected research in this paper was considered to follow the RSM concepts, consisting of different stages (Figure 1). The final 82 selected papers led to the synthesis stages, with additional references to the standard RSM, such as those within [1,19].

**Table 1.** Inclusion criteria for filtering papers.

| Paper Inclusion Criteria | Paper Exclusion Criteria |
| --- | --- |
| Application of observational or historical data as an alternative to the DoE in RSM | The RSM should not conduct a designed experiment to obtain data (however, some papers still referred to nondesigned experiments/non-DoE with a rationale of hard-to-control factors; the details are in Figure 8) |
| Involving previous experimental data for RSM, some papers referred to combined datasets from previous experiments | The RSM entirely refers to the dataset without completing it, with new additional experiments. |
| Involvement of the three stages of standard RSM analysis (DoE, modeling, and optimization) | One of the stages of standard RSM analysis is missing |
| RSM analysis involves searching for influencing factors, similar to the original RSM concept | A direct prediction system with real-time data recording and modeling is not a part of this SLR because no such analysis of significant influencing factors exists. |

## 4. Descriptive and Bibliometric Analysis

The descriptive analysis in this section explained the research trends associated with the topics in this paper, and the bibliometric analysis focused on the methods involved in the RSM-OD and research fields to which it has been applied. Since the early 2000s (Figure 5), the increased number of indexed publications in the SCOPUS database with search strings (Figure 4) shows that the application of the RSM-OD has occurred in various research fields. Table 2 shows some of the research fields where the method has been applied.

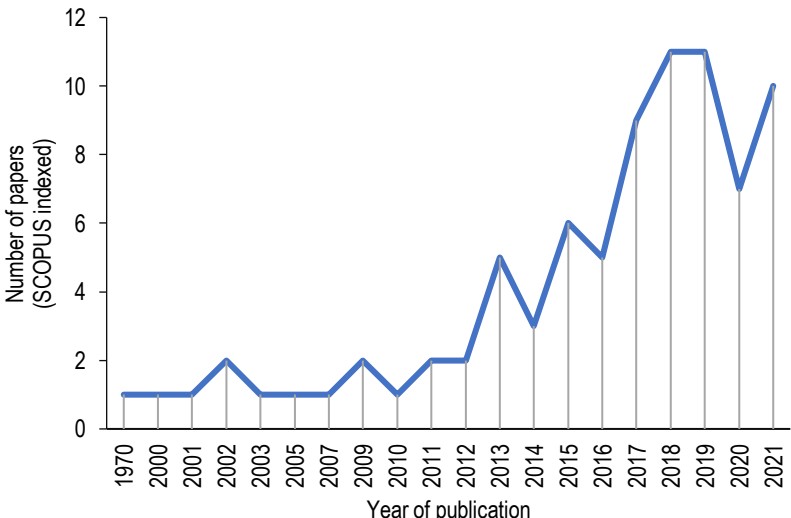

**Figure 5.** Paper trends for RSM-OD.

The pharmacy/chemistry/chemical engineering fields commonly deal with laboratory-scope experiments. They can be improved with the use of standard RSM, but they use already provided data as the input for RSM. Meanwhile, manufacturing, petroleum, and similar engineering fields with modern equipment mostly have a data-acquiring system.

Thus, using the provided dataset rather than experimental data is more reasonable. Furthermore, using a treemap (Figure 6) to categorize the journal quartile shows that the highly impacted journal (Q1/Q2) gives the highest percentage among other quartiles, which means that the RSM-OD has supported high-quality research. For Q1 journals, the research field of (pharmacy/chemistry/chemical) engineering (10.00%) and manufacturing processes (11.25%) still dominated regarding the application of RSM-OD, followed by other fields. A similar interpretation is also drawn for Q2 and the others. Thus, scholars have opportunities to develop RSM-OD procedures required by various research fields involving designed, experiment-based optimization processes at any level of the impacted journals.

**Table 2.** Distribution of papers based on research fields.

| Field of Application of RSM-OD | Percentage |
| --- | --- |
| pharmacy/chemistry/chemical engineering | 22.50% |
| manufacturing process | 18.75% |
| petroleum/coal/mining | 11.25% |
| cleaner production/waste | 10.00% |
| material & mechanical engineering | 7.50% |
| energy | 6.25% |
| food | 5.00% |
| civil engineering | 3.75% |
| medical science | 3.75% |
| aerospace | 2.50% |
| biology | 2.50% |
| methodological development | 2.50% |
| waste processing | 2.50% |
| social science | 1.25% |

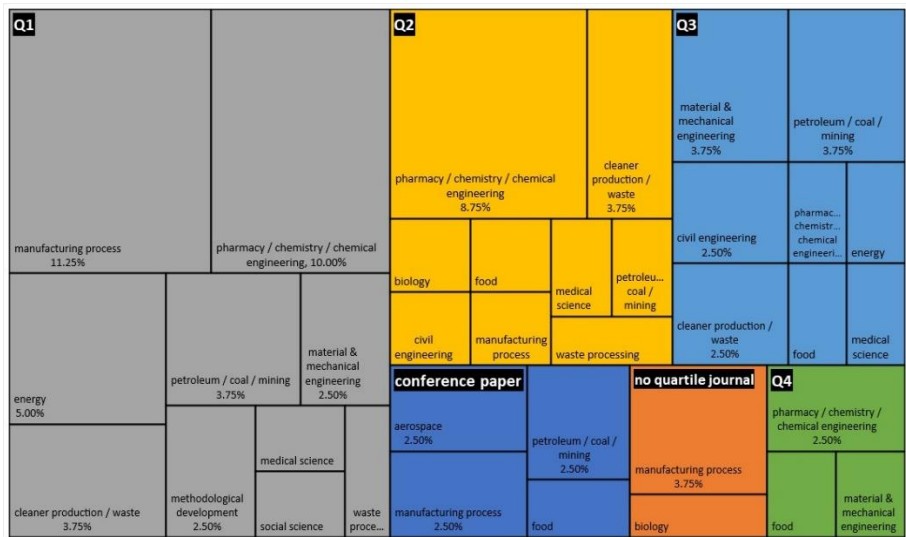

**Figure 6.** Journal quartile by research field.

VOSviewer® v.1.6.15 software provided by Centre for Science and Technology Studies of Leiden University was used to obtain the graphical network in Figure 7. In the figure, the author's keywords represent various terms incorporated in the RSM-OD. The figure also gives insights into the development of the integrated RSM tools/methods to handle nonexperimental data, particularly for certain RSM-related methodological terms, although specific research field-related keywords were still included. The bibliometric analysis consisted of nodes and the links connecting them. Large nodes represent high keyword occurrences, and the links indicate co-occurrences in the same papers. Table 3 shows the results of the complete bibliometric analysis, including the total link strength, which exhibited a high number of co-occurrences between the keywords.

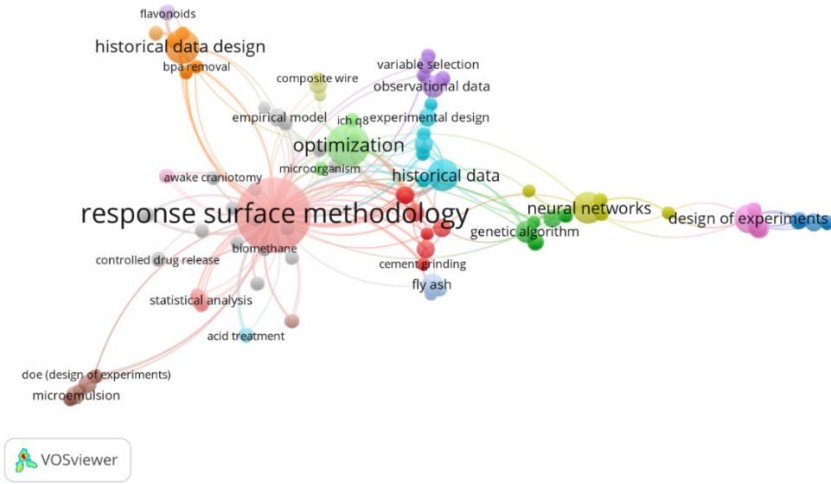

**Figure 7.** Graphical network of bibliometric analysis.

**Table 3.** Occurrences and link strength of graphical keyword networks in Figure 8.

| Author's Selected Methodological Keywords (Excluding Specific Research Field Keywords) | Occurrences | Links | Total Link Strength |
|---|---|---|---|
| RSM | 33 | 130 | 144 |
| optimization | 11 | 42 | 51 |
| HDD only | 7 | 27 | 29 |
| historical data | 6 | 26 | 32 |
| neural networks | 6 | 23 | 24 |
| DoE | 5 | 23 | 27 |
| genetic algorithm | 3 | 15 | 15 |
| observational data | 3 | 13 | 13 |
| Analysis of variance (ANOVA) | 2 | 15 | 16 |
| quality by design | 2 | 14 | 14 |
| modeling | 2 | 9 | 10 |
| statistical analysis | 2 | 9 | 10 |
| Taguchi method | 2 | 9 | 9 |
| process optimization | 2 | 8 | 9 |
| experimental design | 2 | 8 | 8 |
| retrospective data | 2 | 6 | 10 |
| intelligent systems | 1 | 7 | 7 |
| machine learning | 1 | 7 | 7 |
| response-surface designs | 1 | 7 | 7 |
| six sigma | 1 | 7 | 7 |
| support vector machine | 1 | 7 | 7 |
| industrial-scale optimization | 1 | 6 | 6 |
| RSM historical data modeling | 1 | 5 | 5 |
| causality | 1 | 5 | 5 |
| data-driven modeling | 1 | 5 | 5 |
| meta-heuristic optimization | 1 | 5 | 5 |

Note: The red highlighted portion represents common RSM terms, the yellow highlighted part denotes high occurrences, and the blue highlighted section denotes low occurrences in Figure 7.

Keywords from research that applied the standard RSM mainly consisted of common terms in the analysis stages, such as DoE and optimization (Table 3, with red, highlighted rows). By ignoring specific terms related to research fields, only methodological terms are shown in Table 3, including those with high (yellow highlighted) and low occurrences (blue highlighted). The largest cluster with the highest occurrences had "RSM" as the main keyword, followed by "optimization" and "DoE"; these three keywords represent the common terms in classic RSM analysis. Therefore, their high occurrences were expected. The analysis also focused on other clusters supporting them and denotes the development of RSM-OD (the yellow highlighted portion in Table 3).

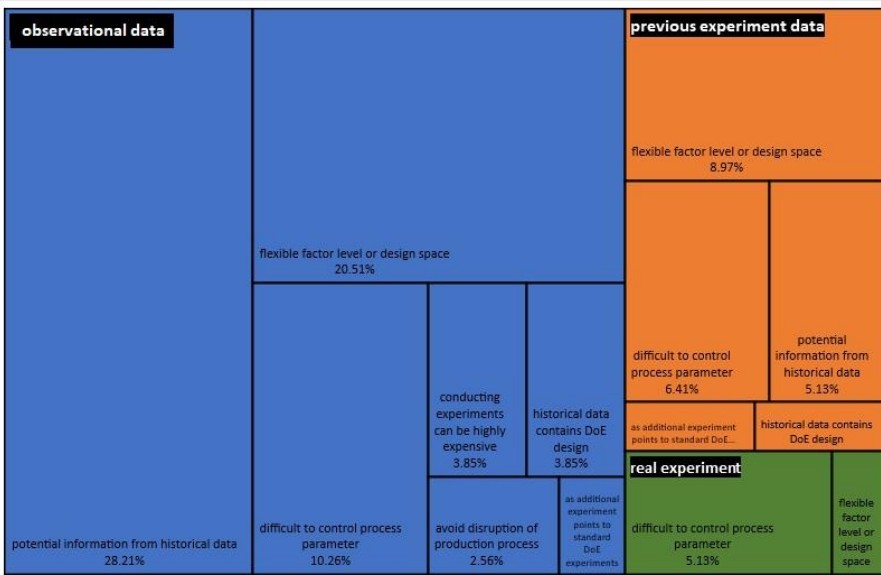

**Figure 8.** Distribution of the rationales based on data types.

The keyword "HDD" gives a high link strength with RSM because it is a term taken from the Design-Expert® v.11 software by Stat-Ease Inc., and the word design is related to a designed experiment based on historical or observational data. The historical word data with similar link strengths were also located near RSM and were strengthened by word observational data, although they showed a low occurrence. Thus, the RSM analysis performed in the papers applied observational/historical data as the input. Subsequently, the keyword "neural networks" formed a cluster near the "genetic algorithm", and these keywords were located alongside the RSM. These keywords corresponded with the RSM model that was replaced by neural networks, and the optimization techniques adopted a genetic algorithm. Furthermore, the blue-highlighted keywords completed the bibliometric analysis, with specific methodological keywords from various research fields. These keywords still showed a relationship with the RSM stages, i.e., DoE, modeling, and optimization, and offered insights into the development of RSM-OD.

## 5. Synthesis and Discussion

The use of observational data in RSM is not without its critics. This practice contradicts the golden standard in RSM and runs considerable risks of being used as an alternative to experimental data. The authors of [38] wrote that using observational data as a replacement for experimental data is risky because of the absence of controllable factors, spurious correlation, and the rise of potential multicollinearity or nonorthogonality. This finding was similar to the problem of semiconductor production in the work of [3] and the slurry thickening process in [39], where observational data contained undetected and uninterpretable multicollinearity, given the application of typical observational-based regression analysis and the need for careful handling ([40]). This opinion was also strengthened by the authors of [19], who wrote a systematic literature review of classic RSM development and showed that orthogonality between factors should be reached to perform individual analyses of each factor. Moreover, the ideal experimentally-based RSM accommodates the procedure of the steepest ascent for the shifting of factor levels in a specific direction toward a stationary optimum response point [1,19]. The use of observational data presents a challenge in conducting this procedure, given the limited range of factor levels. The optimization region is also limited to these available level ranges, as observed in all of the RSM-OD references.

The literature review questions in the previous section served as a guide for the writing order, starting with descriptive and bibliometric analyses. Later, the synthesis stage was performed in line with answering LRQ1 to LRQ3 and continued with the discussion

section. Some treemaps used in this paper simplified the interpretation of the descriptive and research questions answered. Treemaps are used to hierarchically graph structured information, which uses 100% of the available graph space [41], and acted as an excellent application for the supporting systematic literature review in [20].

LRQ1: What are the rationales for using observational data as an alternative to conducting a real RSM experiment?

Approximately 70.51% of the papers employed observational data as the input for RSM, 23.08% were based on previous experimental data, and the remaining 6.41% referred to real experiment data (Figure 8). Observation-based data were obtained depending on the kind of data-acquiring system in the process being studied, and previous-experiment-based data were collected from associated research. The data contained the factor (or X variables) and response (Y variables) with continuous scales, as required by the RSM analysis. The rationale with the highest percentage in Table 4 is potential information that may exist within the observational data. The next highest percentage is the flexible factor level (or design space), where an RSM analysis should be flexible enough to accommodate uncontrolled factor levels within observational data. Moreover, the difficulties in fully controlling the factor levels during a continuous process showed the limitations in conducting designed experiments and provided data that were a better option. This rationale also revealed a high percentage.

**Table 4.** Rationales for selecting RSM-OD.

| Rationales from Papers | Percentage |
| --- | --- |
| potential information from observational data | 33.33% |
| flexible factor level or design space (using the data as provided) | 30.77% |
| difficult to control process parameters | 21.79% |
| historical data contain DoE | 5.13% |
| conducting experiments can be highly expensive | 3.85% |
| additional experiment points to standard DoE experiments | 2.56% |
| avoid disruption to the production process | 2.56% |

Several papers acquired real experiment data but used RSM-OD because of difficulties in controlling the factor levels. They assumed the real experimental data as being observational and argued that modifying the RSM approach based on a nondesigned experiment was easier than conducting a formal standard RSM.

LRQ2: What condition within historical data can be adapted to RSM?

Conducting the DoE experiment ensures the orthogonality between the factors, and the ANOVA can separate each variance for the independent interpretation of their effects ([42]). On the other hand, observational data violate common assumptions in designed experiment data, such as treatment randomization and interfactor orthogonality, as the researcher cannot fully control each factor's level (see an editorial by [38]). Therefore, this section evaluated each reference paper to capture how they explain the condition of data before treating them as the input for RSM modeling based on different approaches in adopting data, i.e., using all observations or obtaining their subsets (Figure 9).

Table 5 shows that more than 70% of papers did not mention specific raw-data conditions. Therefore, they adopted observational data directly as the input for this RSM-OD. The mathematical model and optimization technique were previously determined without evaluating data conditions because they forced the data to fit the model, whether linear or nonlinear, even ignoring the absence of randomization within data. Meanwhile, 5.19% of the papers followed the data condition as it was, which means that the RSM-OD model and optimization techniques were adjusted to adapt to the data condition, and a linear or nonlinear model was selected to give the best fit to the data. A total of 23.38% of the papers explicitly mentioned other conditions, such as factor independencies, ensuring orthogonality, and outlier removal, and considered assumptions as if they were a DoE

experiment. Some papers used orthogonality criteria for evaluating data conditions, such as variance inflation factors (for example, a paper by [43] and a data matrix subsetting used to achieve orthogonality in [3]).

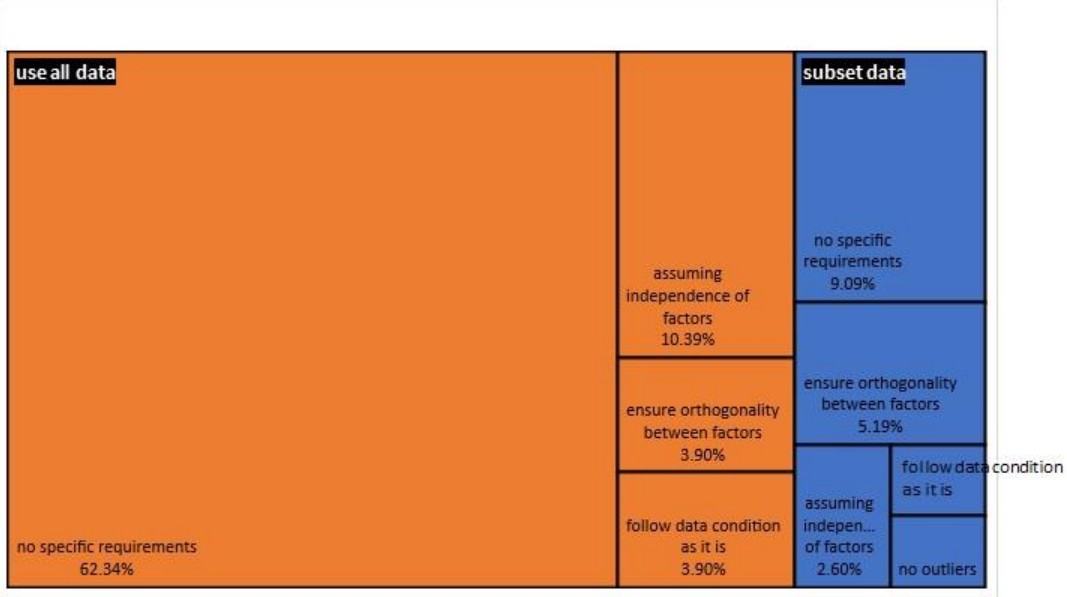

**Figure 9.** Data conditions based on the number of data involved in RSM-OD.

**Table 5.** Required data condition for RSM-OD.

| Observational Data Condition | Percentage |
|---|---|
| No specific data condition requirement (model and optimization stage were determined without considering data condition) | 71.43% |
| Assuming independence of factors | 12.99% |
| Ensure orthogonality between factors | 9.09% |
| Follow data condition as it is (specify RSM-OD model and optimization-based data condition) | 5.19% |
| No outliers | 1.30% |

LRQ3: How historical data are adopted to RSM?

As shown in Figure 1, the three stages of RSM form the integrated procedures for DoE-based optimization. Ideally, the RSM-OD with similar optimization purposes should also adopt these stages. The answers to LRQ2 explain how standard RSM stages with modifications adapt to observational data. Especially at the designed-experiment stage, several approaches show how the RSM-OD treats data as the input to the RSM analysis.

At the DoE stage (Figure 10), two approaches were used to adopt observational data: the first type used all observations (80.52%), and the second type selected a subset (19.48%), with some required conditions. Those that used all provided observations mostly did not need to adopt a DoE. All observations were treated as an input for the RSM model, and the optimum was found based on this input. A few of these papers filtered data to remove unusual observations before RSM modeling. As for the other types, some observations were selected as the subset data for RSM modeling based on specific criteria. Mainly, the requirement of orthogonality between factors was one of the reasons for selecting observations into a subset; these criteria are required in standard RSM analysis and fulfilled by the DoE. Thus, a certain DoE-like adaptation is needed in the RSM-OD analysis, including common assumptions, such as treatment randomization and interfactor independence (orthogonality). Figure 10 shows that standard DoEs, such as factorial, optimal, and Taguchi designs, were used as references in selecting the observation subset.

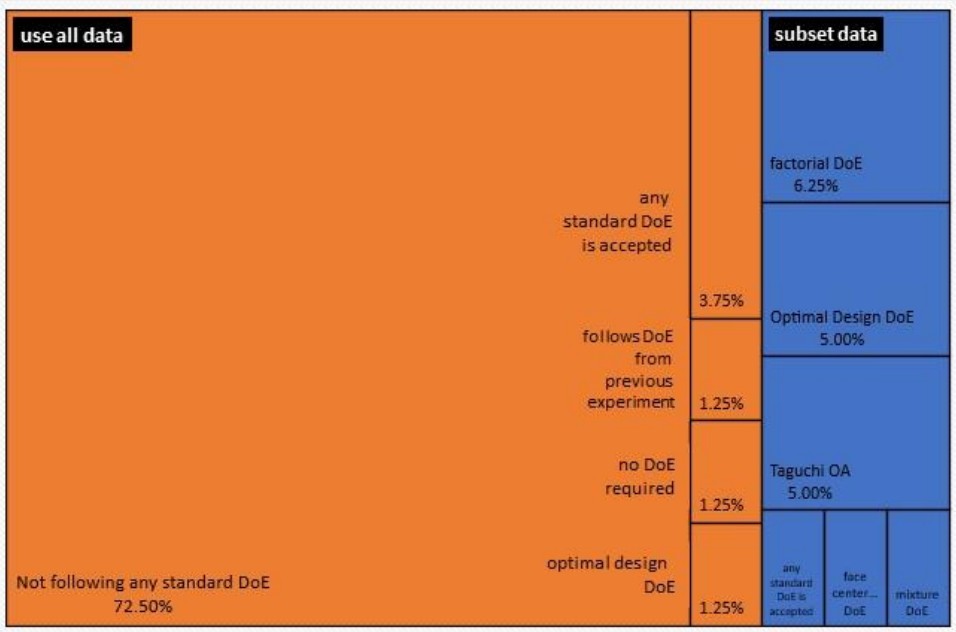

**Figure 10.** DoE stage.

In the modeling stage (Figure 11), almost all papers (90.54%) applied a linear model; the others used a neural network (6.76%), and the rest used other models, such as the Taguchi and support vector model (2.7%). As a common linear model in RSM, this approach works as the standard RSM completed by typical statistical analyses, such as factor significance and R-square. For the neural network approach, most of the papers implemented it for modeling and optimization purposes. As the neural networks are close to a black-box model without any statistical analysis, the authors still performed ANOVA and R-square analysis to evaluate significant factors and show an interpretable result. Alternatively, the Taguchi method approach, which was proposed in [2,11], was also applied based on the typical signal-to-noise ratio in its analysis.

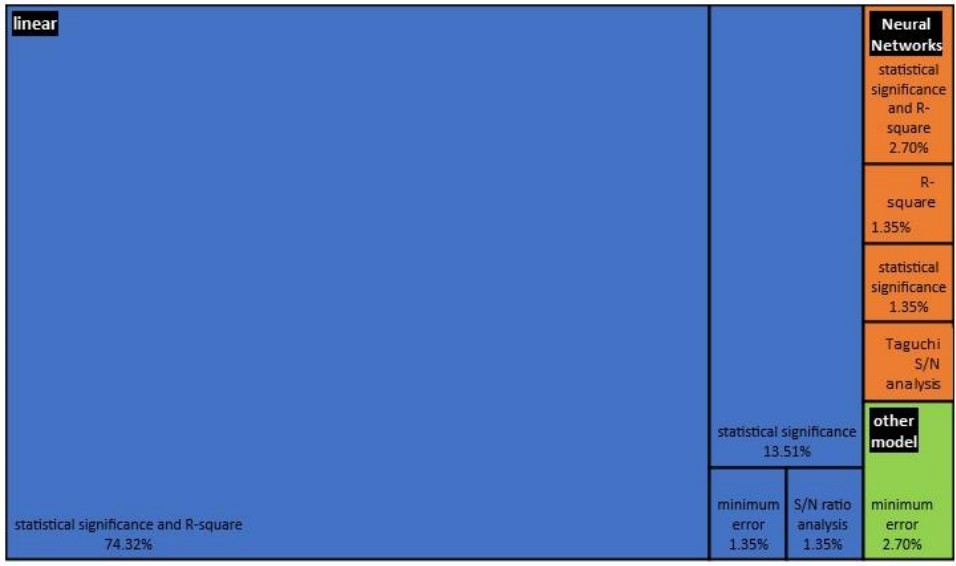

**Figure 11.** RSM modeling stage.

For the optimization stage (Figure 12), as the highest percentage showed a linear model, a standard local search optimization algorithm was preferred and commonly provided

in some software. Moreover, several papers with linear models adopted metaheuristics algorithms to find an optimum response. Notably, the graph in Figure 13 shows that some papers excluded the optimization process, and they only considered the first two stages of RSM-OD for prediction or factor investigation.

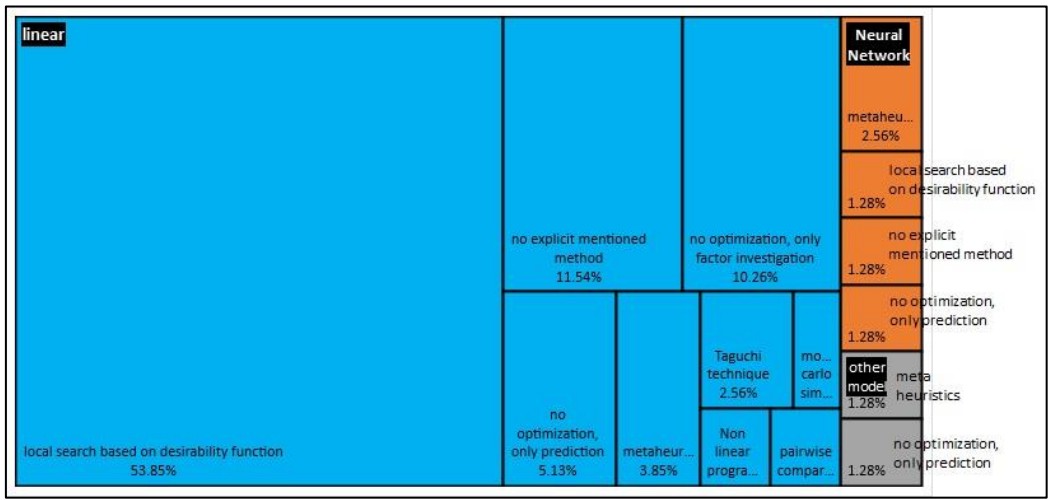

**Figure 12.** Optimization stage.

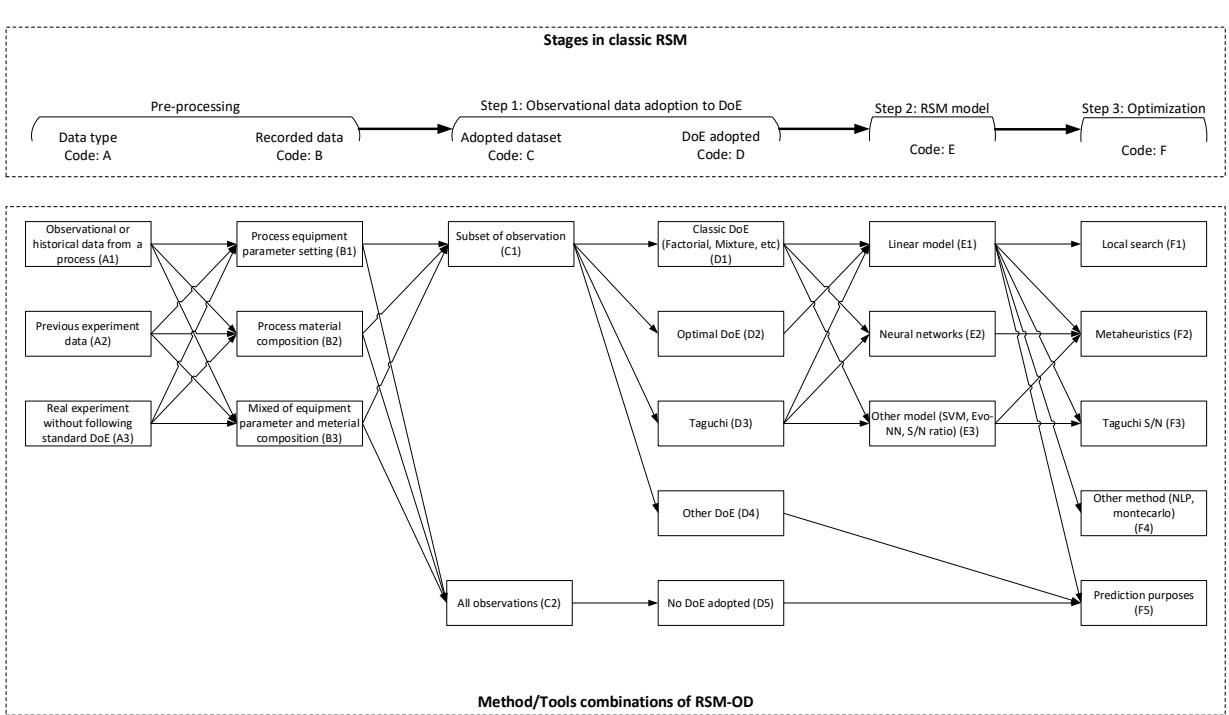

**Figure 13.** Combination of the methods adopted in RSM-OD.

## 5.1. Comparative Analysis

Several approaches to handling observational data for RSM were proposed, and rationales were provided for each based on specific references. Figure 13 represents the combination of tools or methods applied to RSM-OD, and the maps based on the stages in classic RSM analysis are shown in Figure 1. By reading from the left side, each box in the figure represents the tools or methods used in RSM-OD, and the lines denote the other tools/methods at each stage of the RSM analysis. Various modifications in RSM-OD in the reference papers still obeyed the basic principles of classical RSM, and any RSM improvement should not be much different.

The method combinations started with the identification of nondesigned-experimental data (Code A). Several papers referred to observational data from this type of continuous process, and others referred to previous experimental data or conducted an actual nondesigned experiment. Code B categorizes the recorded variables that will be the factors in RSM. Primarily, the studied process records its equipment parameters, the composition of the materials, or both. Code C represents how the provided data will be treated by considering all observations or selecting its subset. Code D categorizes the standard DoE adopted while treating the provided data within the RSM analysis. Code E shows the RSM model adopted, and Code F represents the optimization technique.

The combinations of Code A–F provided many options. However, the main concept of the three-stage RSM became the focus of grouping each paper. As is shown in Table 6, based on Codes C (stage 1: treating data), E (Stage 2: RSM model), and F (Stage 3: optimization), only seven types of approaches, which were represented by seven clusters, were obtained, and the references are shown in Table 6.

**Table 6.** References for Figure 13.

| Clusters | Three Stages of RSM | | | Additional Stage | | | References |
|---|---|---|---|---|---|---|---|
| | Stage 1 (Code C) | Stage 2 (Code E) | Stage 3 (Code F) | Code A | Code B | Code D | |
| Cluster 1: Subset—Linear model—local search (12.05%) | C1 | E1 | F1 | A1 | B1 | D1 | [44] |
| | | | | | | D3 | [2] |
| | | | F2 | | | D2 | [45] |
| | | | | | B2 | D2 | [35] |
| | | | F4 | | | D1 | [46] |
| | | | | | B1 | D3 | [11] |
| | | | | | | D3 | [47] |
| | | | F5 | | | D1 | [3] |
| | | | | | B2 | D2 | [13,48] |
| Cluster 2: Subset—NN model—metaheuristics. (3.61%) | C1 | E2 | F2 | A1 | B2 | D1 | [49] |
| | | | | | | | [50] |
| | | | F5 | | B3 | | [12] |
| Cluster 3: Subset—other models—other purposes. (1.20%) | C1 | E3 | F5 | A1 | B3 | D3 | [51] |
| Cluster 4: All obs.—linear model—local search (55.42%) | C2 | E1 | F1 | A1 | B1 | D5 | [8,43,52–60] |
| | | | | | B2 | | [7,61–65] |
| | | | | | B3 | | [66,67] |
| | | | | A2 | B1 | | [6,68–73] |
| | | | | | B2 | | [74–77] |
| | | | | | B3 | | [9,78–82] |
| | | | | A3 | B1 | | [83,84] |
| | | | | | B2 | | [85] |
| Cluster 5: All obs—linear model—metaheuristics (10.84%) | C2 | E1 | F2 | A1 | B1 | D4 | [86] |
| | | | | | | D5 | [4,87–91] |
| | | | | | B2 | D1 | [92] |
| | | | | | B3 | D4 | [93] |

**Table 6.** *Cont.*

| Clusters | Three Stages of RSM | | | Additional Stage | | | References |
|---|---|---|---|---|---|---|---|
| | Stage 1 (Code C) | Stage 2 (Code E) | Stage 3 (Code F) | Code A | Code B | Code D | |
| Cluster 6: All obs.—linear model—other optimization technique (8.43%) | C2 | E1 | F4 | A3 | B2 | D4 | [2,94] |
| | | | | | B1 | | [95–97] |
| | | | F5 | A1 | B2 | D5 | [98,99] |
| | | | | A3 | B3 | | [100] |
| Cluster 7: All obs.- NN model—metaheuristics (7.23%) | C2 | E2 | F2 | A1 | | | [101] |
| | | | F5 | A2 | B1 | D5 | [102] |

The most preferred was cluster 4, with 55.42% relativity to all the selected reference papers; it used all observations as the input to the linear RSM model and applied the ordinary local search optimization method. It is similar to standard RSM, but risks may arise during the analysis by selecting all observations. Cluster 1, which was similar to Cluster 4, had the second highest value: 12.05%; the difference is that this cluster selected a subset of observations that fulfilled a particular DoE and guaranteed interfactor orthogonality. Next, Cluster 5 (10.84%) was similar to Cluster 4 but replaced the local optimization method with a metaheuristic technique. A more complex RSM model with all observations as the input became the rationale for this replacement. Cluster 6 (8,43%) applied other optimization techniques, such as Taguchi S/N ratio, linear programming, and the Monte Carlo method [2,44,45]. In Clusters 2 and 7, the linear model was replaced with neural networks to handle the nonlinearity of the observational data, all observations, or the subset data. Moreover, such a complicated black-box neural-network model applied the metaheuristics method to find the optimum. Concerning the three stages of the RSM, a summary of the method combinations (Figure 13 and Table 6) is rewritten in Table 7.

**Table 7.** Summary of method combinations in consideration of the three stages of RSM.

| | | Advantages | Disadvantage |
|---|---|---|---|
| Stage 1 RSM | subset | Selecting a subset based on specific criteria increases inter-factor orthogonality | A number of of observations will be excluded from the RSM analysis |
| | all observation | As a potential source of information, all observations will be included in the RSM analysis | potential multicollinearity between factors and the possibility of outlier observations |
| Stage 2 RSM | linear model | strong foundation with clear inference and interpretation | strictly statistical assumptions |
| | Neural-net model | black-box model free of assumptions | no model interpretation and potential garbage-in-garbage-out |
| | other models | Similar to neural networks, the SVM model has no required assumptions, and the Taguchi method works without a pre-specified mathematical model. | |
| Stage 3 RSM | local search | fast iterative algorithm | potential local optimum |
| | metaheuristics | accommodate global optimum | highly depends on initial conditions |
| | other technique | Some papers with prediction purposes exclude optimization techniques; the others involve linear programming and Monte-Carlo. | |

### 5.2. Advantages and Disadvantages of RSM-OD

Based on the synthesis above, the discussion emphasized how the classic RSM concept methodologically adopts nonexperimental data as an alternative to the DoE experiment. The classic RSM has strong scientific references when integrating the three stages of its

analysis (Figure 1), and each stage also gives a clear theoretical basis. Therefore, any development in RSM, including the fulfillment of assumptions during the analysis, should remain within these stages. Thus, the discussion will explore the methods and combinations used in the reference paper (Figure 13).

According to Table 7, those options that combined the methods within the three stages of RSM raised some advantages and limitations. In stage 1 of the RSM, contradictions existed between the selection of all observations or their subsets. One problem relates to interfactor orthogonality and the other deals with the justification of selecting only a subset from several potentially informative observations. In stage 2, different types of RSM models, i.e., linear (or polynomial) or machine-learning type models, provided different modeling approaches with each of their consequences. The powerful and interpretable linear model works with several strict assumptions, whereas the free-assumption machine-learning-based model contains potential over-fitting and is noninterpretable. In stage 3, the ordinary local search algorithm works best for a single-optimum point linear model, whereas the metaheuristics algorithm provides a larger search area with local and global optima.

By referring to the papers needing observational data, RSM can be developed with alternatives to conduct a real experiment. Notably, observational data will not give pieces of information that are as perfect as within the designed experiments because of the assumptions of violations within. However, numerous references in this paper have shown the success of RSM-OD, although some ignored the concept behind the classic RSM. Therefore, a new procedure must be developed for this type of RSM to fulfill all the required assumptions of the standard classic RSM.

### 5.3. Potential Gaps and Future Research

With all the explained descriptive and synthesis analyses, we identified opportunities and gaps in the development of new RSMs in consideration of adopting observational data (Table 8). Stage 1 deals with how the developed procedures work, according to the concept of classic DoE, including the concept of orthogonality and randomization. Stage 2 developments can be improved when considering model interpretation, including factor significance and goodness-of-fit. Stage 3 deals with the capability of finding the global optimum based on the fitted model in Stage 2. All these opportunities are expected to give a stronger theoretical basis for implementing RSM-OD to complete its practical applications, assuring the users regarding its use.

**Table 8.** Opportunities and gaps for further development.

| RSM Stages | Development Opportunities for Future Research | Potential Gaps in References |
|---|---|---|
| Stage 1 | Develop procedures to adopt observational data considering the concept of classic DoE | Procedure development to:<br>1. fulfill factor orthogonality and its evaluation/measurement<br>2. Improve orthogonality of observational data<br>3. handle non-randomized treatment within observational data<br>4. pre-process observational data (cleaning/filtering/subsetting)<br>5. Dividing variation for each factor, similar to ANOVA |
| Stage 2 | Develop an adaptive RSM mathematical model to adapt observational data concerning required assumptions | Model development to:<br>1. accommodate un-designed/unpatterned observational data<br>2. fulfill model-fitting assumptions, or ignore them<br>3. enhance of model interpretability |

**Table 8.** *Cont.*

| RSM Stages | Development Opportunities for Future Research | Potential Gaps in References |
|---|---|---|
| Stage 3 | Develop an optimization algorithm referring to a pre-defined RSM model | Optimization technique to: <br> 1. provide a comprehensive optimum search area <br> 2. avoid local optimum |

## 6. Conclusions

Using observational data within RSM is promising, particularly when data-recording technology (big data) exists. It was found that the main rationales for adopting observational data within RSM are the existence of historical data and avoiding interruptions in continuous production. However, due to the unstructured, highly variable, and serial-correlated nature of the observational data, data modifications prior to use in the RSM is necessary. Therefore, the paper aims to explore the various methods/approaches for incorporating observational data in RSM through a systematic literature review using the PRISMA framework, from which 83 studies were analyzed. Based on the three stages of classic RSM, modifications can be conducted at each stage, i.e., data treatment, modeling, and optimization. With respect to the first stage (data treatment), the modification involves selecting an observation subset or pretreating the data to increase acceptance in the RSM based on specific criteria, such as orthogonality and treatment randomization. In the second stage, adaptive RSM mathematical models are selected to handle nonideal observational data. Complex nonlinear machine learning models are common approaches for adapting RSM models, for example, the neural network and SVM models. In the last stage, an alternative optimization method suitable for such a complex RSM model is also highlighted. Metaheuristic optimization techniques perform well when finding the optimal factor levels modeled using a nonlinear RSM model. The combinations of the proposed methods for the RSM stages reveal insights into the fact that there is an open potential for developments in RSM-OD as an alternative to classic RSM.

Despite the deviation from standard RSM techniques, the proposed RSM-OD methods in the literature can still achieve their design/process optimization purpose with reasonable results. However, the methods also raised some limitations, such as data orthogonality issues, statistical assumptions, model specifications, model interpretability, and the need for advanced optimization methods.

This paper contributes to the RSM literature by providing the advantages or disadvantages of using observational data for process/design optimization, demonstrating opportunities to further improve the proposed methods in RSM-OD, and coping with their theoretical limitations and unexpressed assumptions. Once those issues are well addressed, RSM-OD may be a promising alternative to classic RSM.

**Author Contributions:** Conceptualization, M.A.H.; methodology, M.A.H., B.M.S. and B.S.W.; formal analysis, M.A.H., B.M.S. and B.S.W.; writing—original draft preparation, M.A.H.; writing—review and editing, B.M.S. and B.S.W.; supervision, B.M.S. and B.S.W.; funding acquisition, M.A.H. All authors have read and agreed to the published version of the manuscript.

**Funding:** This research was funded by Universitas Surabaya (Contract No. 1186/PKD-SL/SDM/IX/2020).

**Institutional Review Board Statement:** Not applicable.

**Informed Consent Statement:** Not applicable.

**Data Availability Statement:** Not applicable.

**Conflicts of Interest:** The authors declare no conflict of interest.

### Abbreviations

| Abbreviations (Alphabetical Order) | Full Form |
| --- | --- |
| DoE | Design of experiment |
| HDD | Historical data design |
| LRQ | Literature review questions |
| NN | Neural network model |
| PRISMA | Preferred reporting items for systematic reviews and meta analyses |
| RSM | Response surface methodology |
| RSM-OD | Observational data-based RSM |
| SLR | Systematic literature review |
| SVM | Support vector machine model |

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
