# Peer review of "Response Surface Methodology Using Observational Data: A Systematic Literature Review"

_applsci, doi:10.3390/app122010663_

Round 1

Reviewer 1 Report

This paper provides a systematic review of existing academic literature to portray the proposed methods used in previous studies to incorporate observational data in response surface methodology analysis. The authors carefully reviewed 83 selected papers for analysis. This paper may be interesting, but it does not work for me. The main concern is that the motivation is unclear, and the conclusion is so weak.  Some comments are listed as follows. 

1. The keywords should be listed in alphabetical order.

2. The introduction is not clear, the author should be added more references in the introduction.

3. Page 3 L 102, the orthogonality is not the only goal for DOE.

4. Page 4 L 125 to 138, this paragraph needs to strengthen. ``prediction" should be ``prediction."

5. In Section 2.2, What is your conclusion in this section? You can give a small conclusion in each section. 

6. Your method or your result is not interesting. It is not a research paper. 

Reviewer 2 Report

The authors performed a systematic review of changes in the classic RSM to incorporate observational data, ie, unplanned experimental data. This is a subject of interest, given the enormous amount of observational data currently generated. Please find my suggestions below.

-English review

-maybe "big data" can also be cited as motivation in th introduction 

-give the meaning of SLR in figure 2

-sections are sometimes repetitive, eg section 3.1; in that sense, I think the manuscript can be restructured and shortened

-if not wrong, only the Scopus database was used; if yes, justify in the manuscript

-please check the order of figures, for example figure 4

-define treemap the first time it is used

-check the correspondence of tables (and figures) between the citations throughout the text and the tables themselves (figures)

-if not wrong, "randomization" appears only at the end of the manuscript (section 5.2); I think this should be discussed early on along with orthogonality; I think "serial correlation" should also be discussed, considering the continuous process as one of the main sources of observational data

Reviewer 3 Report

In the review paper, authors discuss about using observational data in response surface methodology. The review topic is intriguing and promising in the area. Overall, the manuscript structure and content are suitable for the journal Applied Sciences. I am pleased to send you major-level comments, there are some flaws that need to be corrected before publication.

Please consider these suggestions as listed below.

1.     The importance of the current review must be highlighted more in the abstract section.

2.     Authors should justify how this review is important as compared to other review papers on a similar topic.

3.     Authors should merge citations when is cited more than one citation according to the template. Please revise your paper accordingly since some issue occurs in several spots in the paper.

4.     Please revise the order of the figure’s appearance.

5.     Authors should improve the resolution of Figures 7, 9, 4, 5, 6, and 7 on page 16.

6.     The list of abbreviations should be added in the revised version.

7.     Please unify the style of Tables, text alignment, etc.

8.     Please revise the entire manuscript according to instructions provided in the template (chapter numbering issue, etc.)

Round 2

Reviewer 1 Report

The present form is good for me. The aurthor should be check the English. 

Reviewer 3 Report

All suggestions and comments were accepted or answered by the authors. Hence, the manuscript can be accepted for publication.